# Specific ablation of the NCoR corepressor δ splice variant reveals alternative RNA splicing as a key regulator of hepatic metabolism

**Michael L. Goodson**[1¤a]\*, **Trina A. Knotts**[2], **Elsie L. Campbell**[1], **Chelsea A. Snyder**[1¤b], **Briana M. Young**[1¤c], **Martin L. Privalsky**[1]

1 Department of Microbiology and Molecular Genetics, College of Biological Sciences, University of California at Davis, Davis, California, United States of America, 2 Department of Molecular Biosciences, School of Veterinary Medicine and Mouse Metabolic Phenotyping Center, Microbiome & Host Response Core, University of California at Davis, Davis, California, United States of America

¤a Current address: Department of Anatomy, Physiology, and Cell Biology, School of Veterinary Medicine and Mouse Metabolic Phenotyping Center, Microbiome & Host Response Core, University of California at Davis, Davis, California, United States of America
¤b Current address: Gilead Sciences, Foster City, California, United States of America
¤c Current address: Department of Microbiology and Immunology, School of Medicine, University of California at Davis, Davis, California, United States of America
\* mlgoodson@ucdavis.edu

**Data Availability Statement:** All RNAseq Data is available at NCBI GEO (accession# GSE157761).

## Abstract

The NCoR corepressor plays critical roles in mediating transcriptional repression by both nuclear receptors and non-receptor transcription factors. Alternative mRNA splicing of NCoR produces a series of variants with differing molecular and biological properties. The NCoRω splice-variant inhibits adipogenesis whereas the NCoRδ splice-variant promotes it, and mice bearing a splice-specific knockout of NCoRω display enhanced hepatic steatosis and overall weight gain on a high fat diet as well as a greatly increased resistance to diet-induced glucose intolerance. We report here that the reciprocal NCoRδ splice-specific knock-out mice display the contrary phenotypes of reduced hepatic steatosis and reduced weight gain relative to the NCoRω-/- mice. The NCoRδ-/- mice also fail to demonstrate the strong resistance to diet-induced glucose intolerance exhibited by the NCoRω-/- animals. The NCoR δ and ω variants possess both unique and shared transcriptional targets, with expression of certain hepatic genes affected in opposite directions in the two mutants, others altered in one but not the other genotype, and yet others changed in parallel in both NCoRδ-/- and NCoRω-/- animals versus WT. Gene set expression analysis (GSEA) identified a series of lipid, carbohydrate, and amino acid metabolic pathways that are likely to contribute to their distinct steatosis and glucose tolerance phenotypes. We conclude that alternative-splicing of the NCoR corepressor plays a key role in the regulation of hepatic energy storage and utilization, with the NCoRδ and NCoRω variants exerting both opposing and shared functions in many aspects of this phenomenon and in the organism as a whole.

Other relevant data are within the manuscript and its Supporting Information files.

**Funding:** MLP: Funded in part by Public Health Services/National Institutes of Health award NIDDK52528 and American Diabetes Association award ADA712BS151. CAS: Funded in part by Public Health Services/National Institutes of Health NIEHST32-ES007059. Funding of the UC Davis Mouse Biology Program was in part through Public Health Services/National Institutes of Health Grant U24 DK09293. The funders had no role in study design, data collection and analysis, decision to publish, or preparation of the manuscript.

**Competing interests:** The authors have declared that no competing interests exist.

## Introduction

NCoR (Nuclear Co-Repressor; encoded by NCOR1 in humans/Ncor1 in mice) and SMRT (Silencing Mediator of Retinoic Acid Receptor; encoded by NCOR2 in humans/Ncor2 in mice) are auxiliary proteins that interact with a variety of nuclear receptor and non-receptor transcriptional factors to repress gene expression [1–5]. Both function as bridging proteins that bind to their nuclear receptor partners through a series of receptor interaction domains (RIDs) and to various effector proteins through a series of silencing domains (SDs) (Fig 1A) [1–3, 6–9]. NCoR and SMRT play crucial roles in energy metabolism, endocrinology, general physiology, and development; disruptions in corepressor function are associated with a wide variety of human and non-human pathologies [1, 2, 4, 5, 10–22].

Both NCoR and SMRT are expressed as alternatively-spliced variants that differ in relative abundance in different cell types, possess different molecular architectures, and exhibit different biochemical and biological properties [23–33]. NCoR alternative-splicing changes during pre-adipocyte differentiation from expressing NCoRω, a splice-variant that inhibits adipogenesis, to expressing NCoRδ, a splice-variant that stimulates it [34]. NCoRω contains three C-terminal RIDs, NCoRδ only two (Fig 1A and 1B). As a consequence NCoRω and NCoRδ exhibit different affinities for different nuclear receptor partners and mediate distinct effects on gene expression [25, 26, 30, 34]

Consistent with these findings *in vitro*, ablation of expression of NCoRω in mice (while retaining expression of NCoRδ) increased hepatic steatosis, increased weight gain, increased adiposity, and, paradoxically, improved glucose tolerance on a high fat diet (HFD) compared to wild type (WT) controls [35]. The effects of this NCoRω splice-specific knockout were readily distinguishable from those of "pan-splice" knockouts of all NCoR splice variants [35]. We concluded that the NCoRω splice-variant plays unique roles in WT mouse metabolism and suggested that alternative-splicing of NCoR serves as a means by which different cell types customize corepressor function for different purposes.

We report here an analysis of mice engineered to be unable to express the reciprocal splice-variant NCoRδ while retaining NCoRω expression. These NCoRδ-/- mice exhibited greatly reduced hepatic steatosis as well as reduced overall weight gain on an HFD compared to the NCoRω-/- mice. The NCoRδ-/- mice also fail to demonstrate the strong resistance to diet-induced glucose intolerance exhibited by the NCoRω-/- animals. RNA sequence analysis revealed that the NCoR δ and ω variants possess both unique and shared transcriptional targets. GSEA analysis identified a wide variety of lipid, carbohydrate, and amino acid metabolic pathways that are altered in the livers of the NCoRδ-/- versus WT mice, many of which also differ relative to the NCoRω-/- animals and which likely contribute to the distinct steatosis phenotypes of these different genotypes. Our results extend the finding that alternative-splicing of the NCoR corepressor plays a key role in the proper regulation of metabolism in the liver (and the organism as a whole), with the NCoRδ and NCoRω variants able to mediate shared, distinct, and even opposing roles in important aspects of these phenomena.

## Materials and methods

### Generation of NCoRδ splice-specific knockout mice

This study was carried out in strict accordance with the recommendations in the Guide for the Care and Use of Laboratory Animals of the National Institutes of Health and AAALAC guidelines; it was approved by the University of California-Davis Animal Welfare committee as protocol # 17036.

Knockin mice were generated at the University of California, Davis, Mouse Biology Program (MBP). A targeting vector was created by standard recombinant DNA techniques

## A.

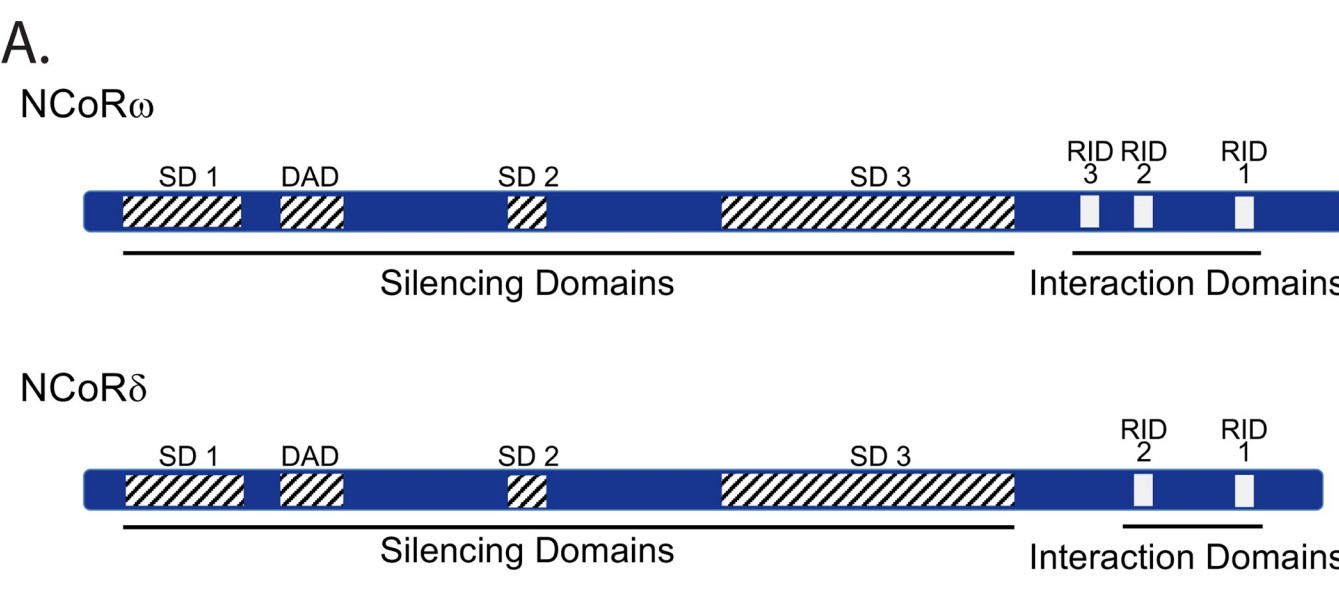

## B.

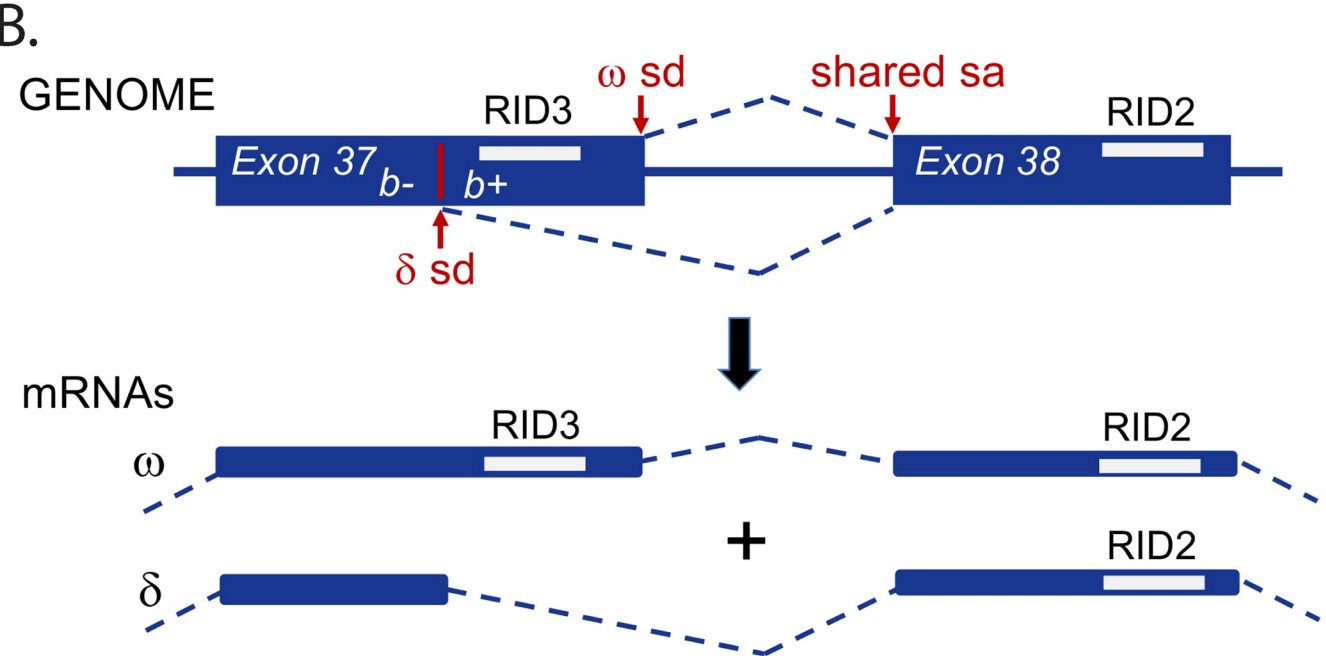

**Fig 1. NCoR alternatively-splice variants.** (A) Schematic comparison of NCoRω and NCoRδ. Locations of Silencing Domains (SD-1 to SD-3 and DAD) and nuclear Receptor Interaction Domains (RID 1 to 3) are shown. (B) Alternative mRNA splicing at the genomic and mRNA levels. The splice donor (sd) and splice acceptor (sa) sites for the NCoRδ and NCoRω variants are indicated.

containing a 16.7-kb fragment of the murine *NCoR* gene (from exons 34 to 41 and the intervening introns) harboring an AGT to TCC mutation that ablates the 5' splice-donor site required for NCoRδ splicing but fully preserves the wild-type amino acid sequence of the NCoRω splice variant (S1A Fig in S1 File). The construct was electroporated into C57BL/6 embryonic stem cells (JM8.F6). The stem cells were genotyped to confirm a single homologous integration of the targeted mutation. Cells were then electroporated with a FLP flippase vector

to remove the FRT-flanked *neomycin resistance* (*neoR*) cassette that was introduced within the intron downstream from exon 37, removing approximately 2.1 kb of non-coding intronic sequence between exons 37 and 38. Clones were again genotyped to verify removal of the neoR cassette and the absence of undesired integrations or rearrangements. Mouse blastocysts (BALBc) were injected with the manipulated ES cells and were implanted in pseudopregnant female mice. Chimeric progeny were then crossed with WT C57BL/6N mice (Taconic Farms, Rensselaer NY). After initial screening by coat color, DNA from tail snips was genotyped by PCR using primers that selectively recognize WT, NCoRδ-/- or NCoRω-/- unique sequences [35]. Animals were housed at up to four mice per ventilated isolator cages with Envigo/Teklad #7092 corncob bedding and Shepherd Specialty Papers Enviro-dri enrichment. Mice were maintained in a 12 h light/dark cycle at approximately 22°C with *ad libitum* access to water and regular mouse chow (Purina 5058, PicoLab MouseDiet 20, Woodstock Ontario). All interventions were during the light cycle. Age-matched male mice generated from our breeding colonies were used for all experiments.

## High/Low fat feeding studies

Mice were maintained on either a HFD (60 kcal% fat; Research Diets D12492, New Brunswick NJ) or a low-fat diet, LFD (10 kcal% fat; Research Diets D12450B) from 8 to 26 weeks with food and water *ad libitum*. Mice were weighed weekly.

## Glucose and insulin tolerance tests

Mice from HFD and LFD feeding studies were fasted for 6 h and weighed prior to an intraperitoneal administration of dextrose (2 g/kg) or Novolin-R insulin (2 U/kg, Novo-Nordisk, Princeton NJ) in sterile saline. Blood glucose was sampled from a tail clip at zero time (i.e. prior to glucose or insulin administration) and at intervals thereafter and measured using a TRUEresult glucometer (Nipro Diagnostics, Fort Lauderdale, FL).

## Hepatic triglyceride analysis

Liver triglyceride analysis was performed by the UC Davis Mouse Metabolic Phenotyping Center (cat.# D3302). Mice were fasted 6 h. prior to termination and the livers were excised and immediately frozen in liquid nitrogen. Weighed liver samples were homogenized in methanol:chloroform, extracted overnight, 0.7% sodium chloride added, and duplicate aliquots of the chloroform/lipid layer were dried under nitrogen gas. Lipid was solubilized in isopropyl alcohol and assayed for total triglycerides spectrophotometrically using enzymatic reagents from Fisher Diagnostics (Middletown, VA).

## GST pulldown assays

In vitro "GST pulldown" protein-interaction assays were performed as previously described [36]. Glutathione S-transferase (GST) or GST-fusion proteins with the receptor interaction domains (RID) of NCoRω and NCoRδ (amino acids 1817–2453 of NCoRω (GenBank accession number U35312) or equivalent region of NCoRδ) were expressed in Sf9 cells using the Bac-to-Bac baculovirus expression system (Invitrogen, Carlsbad, CA). Whole cell extracts were prepared from ~3 x $10^8$ infected Sf9 cells grown in 50 ml of suspension culture using Insect-XPRESS (Lonza, Walkersville, MD) medium containing 9% fetal bovine serum. Three days after infection, Sf9 cells were collected, rapidly frozen in liquid nitrogen, and thawed on ice before resuspending in WCE buffer (20 mM HEPES (pH 7.9), 200mM KCl, 1mM EDTA, 0.5% IGEPAL CA-630, 10% glycerol, 1mM dithiothreitol, 1x Complete Protease Inhibitor

mixture (Roche Molecular Biochemicals, Indianapolis, IN). Resuspended cells were again frozen in liquid nitrogen and thawed on ice. Lysates were cleared by centrifuging at 14,000 x g for 5 min at 4˚C. Cleared lysates were immobilized by binding to glutathione-agarose (Sigma) at 4˚C for 90 min with rocking. The immobilized GST-fusion proteins were pelleted, washed three times with PBS, 0.1% Tween 20 prior to incubation with radiolabeled nuclear receptor proteins.

pSG5-plasmid constructs containing full length nuclear receptors were transcribed and translated into $^{35}$S-radiolabeled proteins *in vitro* using a T7 RNA polymerase-coupled TnT Quick kit (Promega Corp., Madison WI). Each radiolabeled protein (typically 2–5 μl of TnT reaction product per assay) was subsequently incubated at 4˚C with the immobilized GST fusion protein (∼50 ng of GST fusion protein immobilized to ~10 μl of agarose matrix per reaction) in a total volume of 120 μl of HEMG buffer [4 mM HEPES (pH 7.8), 0.2 mM EDTA, 5 mM MgCL2, 10% glycerol, 100 mM KCl, 0.1% Nonidet P-40, and 1.5 mM dithiothreitol] containing 10 mg/ml of BSA and 1× Complete Proteinase Inhibitor. The binding reactions were performed in 96-well Multiscreen filter plates (Millipore, Bedford, MA) placed on a rotating platform to ensure constant mixing. After a 3 hr incubation at 4˚C, the filter wells were washed four times with 200 μl of ice-cold HEMG buffer each, and any radiolabeled proteins remaining bound to the immobilized GST fusion proteins were subsequently eluted with 50 μl of 10 mM glutathione in 50 mM Tris-HCl (pH 7.8). The eluted proteins were resolved by SDS-PAGE and were visualized and quantified using a PhosphorImager/STORM system (Molecular Dynamics, Sunnyvale, CA)

## RT-PCR analysis of gene expression and mRNA splicing

Total RNA was isolated using Qiagen's RNeasy Plus Kit and the manufacturers protocol (Qiagen, Hilden Germany). Complimentary DNA (cDNA) was prepared using the High-Capacity cDNA Reverse Transcription Kit from 1μg of total RNA according to the manufacturers protocol (Thermo Fisher Scientific, Waltham, MA). Quantitative RT-PCR was performed using Powerup SYBR Green Master Mix and a Quantstudio 6 Flex quantitative thermal cycler (Thermo Fisher Scientific, Waltham, MA). Threshold (Ct) analysis was performed using Quantstudio Real Time PCR software (version 1.3). Relative expression was calculated using -ΔΔCt versus the mean of HFD fed WT mice. Analysis of NCoR exon 37b splicing was done by conventional PCR for 28 cycles with an annealing temperature of 62˚C. PCR products were resolved on a 1.5% TAE agarose gel and visualized using SYBR Safe DNA Gel Stain (Thermo Fisher Scientific, Waltham, MA). Oligonucleotide sequences used are described S2 Table.

## RNAseq analysis of gene expression

Total RNA was isolated using Qiagen's RNeasy Plus Kit and the manufacturers protocol (Qiagen, Hilden Germany). Complimentary DNA (cDNA) libraries were prepared with a TruSeq RNA Library Prep Kit (Illumina San Diego, CA). Libraries were sequenced using an Illumina HiSeq3000, generating 1.4–2.5 x$10^7$ 50-base sequence reads per library. Preliminary quality metrics were assessed using FastQC (version 0.11.8, Babraham Bioinformatics, Cambridge, UK). Sequences were aligned to the Genome Reference Consortium Mouse Build 38 (mm10) cDNA Transcriptome and quantified using Salmon [37]. Differential Gene Expression (DGE) analysis was performed using R (version 3.6.1, [38]) and the Bioconductor package DESeq2 (version 1.24.0, [38, 39]. All RNAseq data analyzed in this manuscript is available at GEO accession number GSE157761. Pathway analysis was performed using Gene Set Enrichment Analysis (GSEA) software [40, 41].

## Data analysis

Data analysis was performed using R (version 3.6.1) and RStudio (version 1.2, Rstudio, Boston, MA). Graphs were prepared using either the R package ggplot2 (version 3.2.1, [42] or Prism (version 8.2, GraphPad Software, San Diego CA). Statistical significance was determined using non-parametric 1-way or 2-way ANOVA followed by Dunn's or Tukey Multiple Comparison Tests, as appropriate. For the analysis of differential gene expression by RNA-seq analysis, statistical significance was determined using negative binomial GLM/Wald's significance test (DESeq2), the Benjamini-Hochberg adjustment method, and a padj < 0.05 cutoff.

## Results

### The NCoRδ-/- mice displayed reduced levels of hepatic steatosis and reduced overall weight gain compared to NCoRω-/- mice

WT mice undergo alternative mRNA splicing that incorporates either exon 37b+ or exon 37b- into the expressed NCoR mRNA, producing the NCoRω and NCoRδ proteins respectively (Fig 1A and 1B). NCoRω and NCoRδ include or lack, respectively, RID3, a corepressor domain that physically contributes to, and determines the specificity of, the interactions these corepressors make with their transcription factor partners (Fig 1A and 1B; [25, 30, 34, 43–46]).

Mice genetically manipulated to express NCoRδ but not NCoRω (i.e. NCoRω-/-) exhibit greatly elevated hepatic steatosis and an elevated overall weight gain on a HFD regimen compared to WT [35]. To investigate the consequences of the reciprocal knockout we generated C57BL/6 mice that abolish expression of NCoRδ (i.e. NCoRδ-/-) but that do not inhibit expression or alter the amino acid sequence of the encoded NCoRω protein (S1A Fig in S1 File). NCoRδ-/- mice were recovered at the expected Mendelian ratios, were fully viable and fertile, and displayed no overt anatomical differences compared to WT when maintained on standard mouse chow. RNA analysis confirmed that NCoRδ was undetectable in the tissues examined and total levels of NCoR mRNA expression were not altered, but as anticipated expression was diverted from the mix of NCoRω and NCoRδ seen in WT to the sole expression of NCoRω (S1B Fig in S1 File). The NCoRδ and NCoRω proteins cannot be resolved by immunoblot due to their very similar large sizes and NCoRδ (differing from NCoRω only due to the absence of exon 37b+ sequences) lacks unique epitopes for raising NCoRδ-specific antisera. Nonetheless, given the undetectable levels of NCoRδ mRNA in the knockout, it is unlikely that NCoRδ protein is expressed at any significant level in these animals.

There were several modest perturbations in the hematology of the NCoRδ-/- mice versus WT animals (e.g. an increase in neutrophil counts compared to WT on the LFD versus a decrease on the HFD; S2 Fig in S1 File). The physiological implications of these hematological changes were not further explored.

The NCoRδ-/- animals on the HFD also gained weight at a reduced rate compared to WT animals and at a still further reduced rate compared to the NCoRω-/- mice (Fig 2A), whereas both the NCoRδ-/- and NCoRω-/- mice gained weight more slowly than WT on the LFD (Fig 2A). The NCoRδ-/- mice also did not manifest the enhanced visceral adiposity seen in the NCoRω-/- mice compared to WT animals, although the epididymal adipocytes of both mutants displayed larger median sizes than did the WT adipocytes (S3 Fig in S1 File). Food consumption by the NCoRδ-/- and NCoRω-/- mice was slightly reduced compared to WT on the HFD; the reverse was seen on the LFD (Fig 2B).

Significantly the NCoRδ-/- animals exhibited a greatly reduced hepatic steatosis compared to the NCoRω-/- genotype on the HFD. This was manifested as lower liver weights, as reduced liver triglyceride accumulation, and in histological visualizations (Fig 3A–3C). No statistically

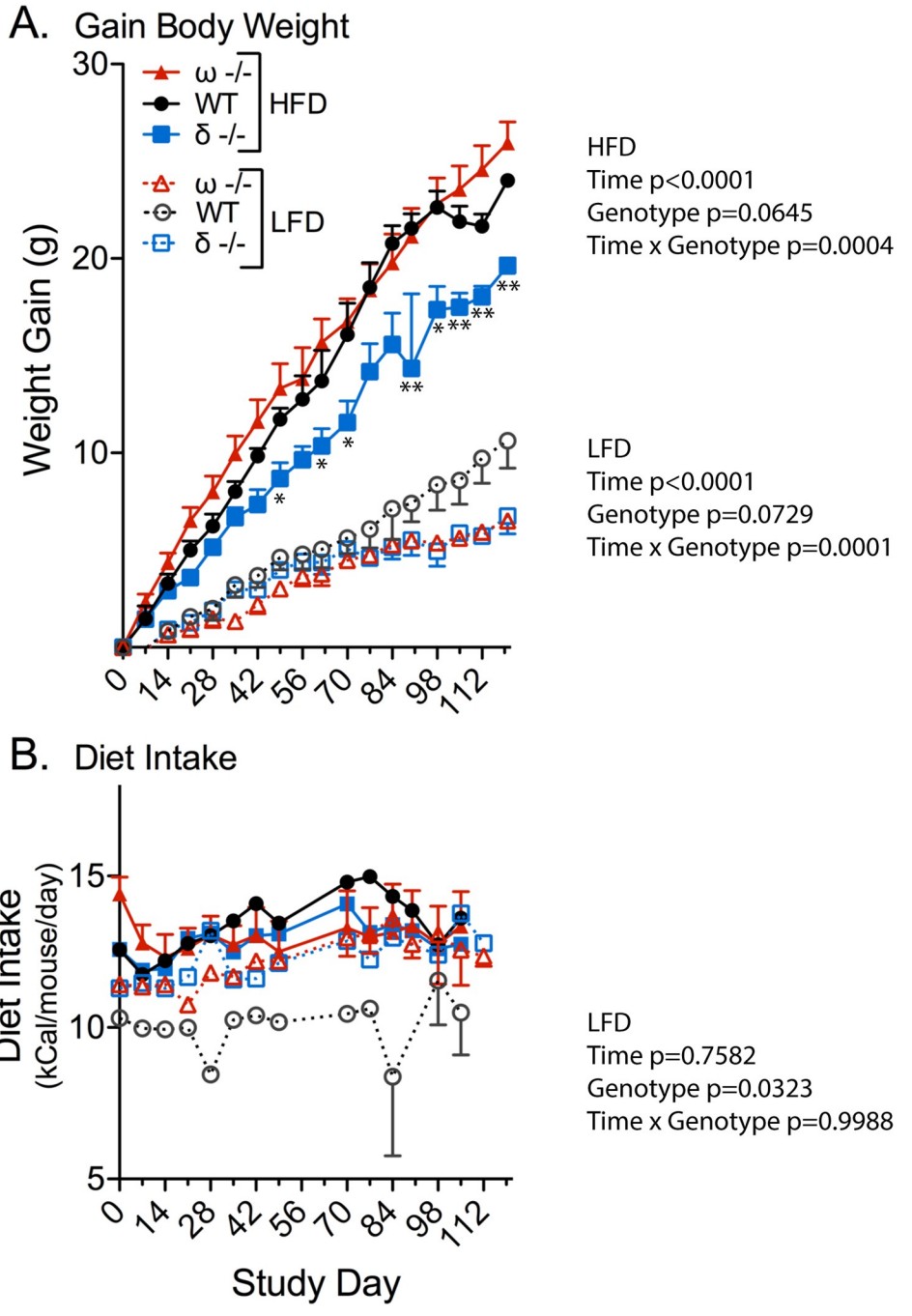

**Fig 2. Weight gain/food intake by NCoRδ-/-, NCoRω-/-, and WT mice.** (A) Gain in body weight versus day on LFD or HFD diet. (B) Dietary food intake. Symbols are as in panel A. Standard errors are indicated as bars. A two-way repeated measure mixed-effect model follow by Tukey post hoc analysis was used to determine the significance of time, genotype, or the interaction. n = 3–4, except for HFD fed p NCoRω-/- mice (n = 12); p-values are indicated.

significant differences were observed in comparisons of the NCoRδ-/- to WT mice (Fig 3). We conclude that alternative splicing of NCoR exon 37 in the WT organism plays an important role in hepatic lipid deposition, with the continued expression of the NCoRω splice-variant in the NCoRδ-/- mice sufficient to prevent the severe hepatic steatosis associated with its ablation in the NCoRω-/- animals.

## A. Liver Weight

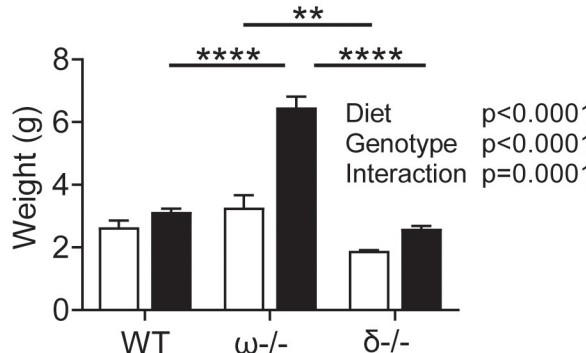

## B. Liver Triglycerides

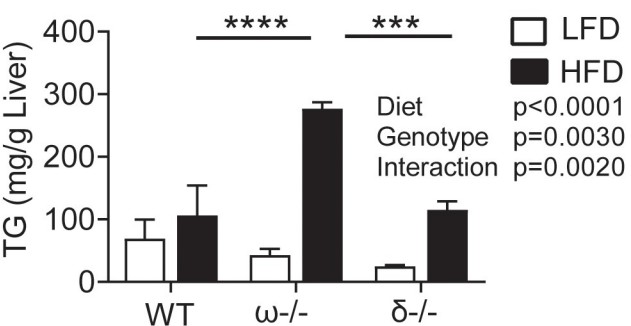

## C. H & E Staining (HFD)

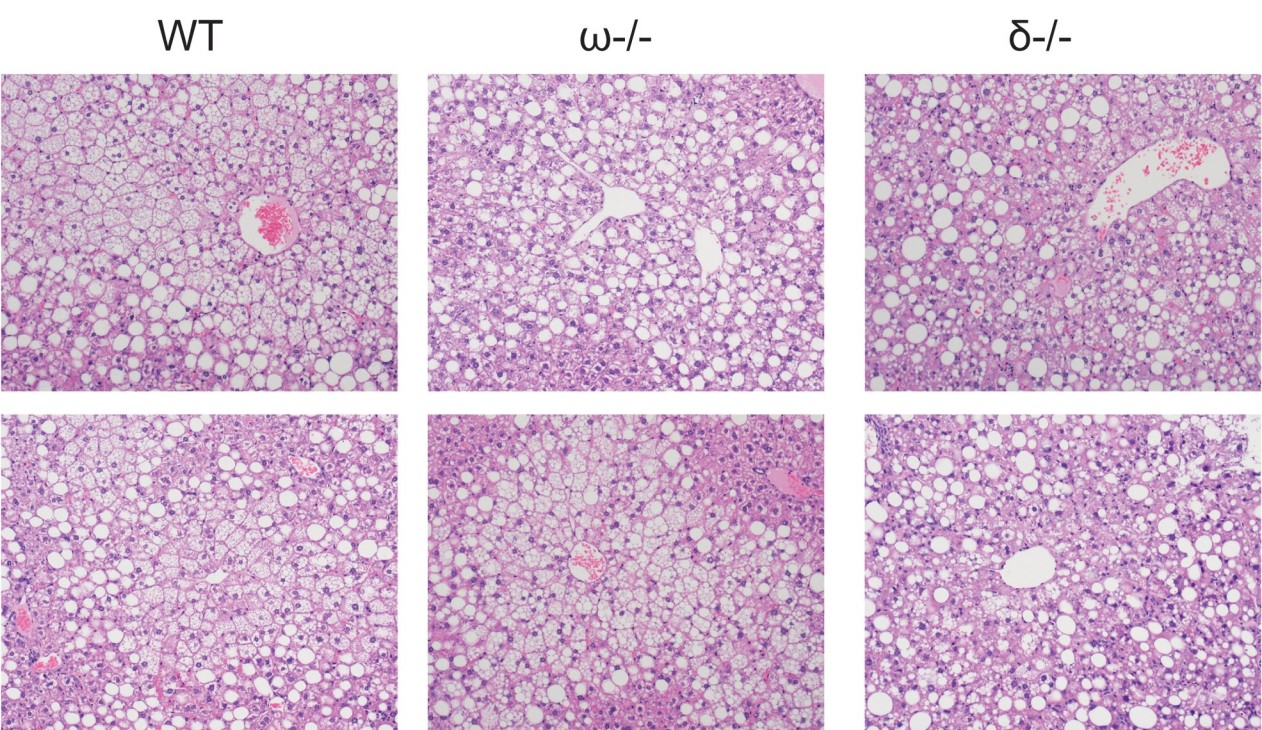

**Fig 3. Liver steatosis in NCoRδ-/-, NCoRω-/-, and WT mice.** (A) Liver weights. (B) Liver triglycerides. Symbol keys are to the right of the panels. Means and standard errors are indicated with statistical significance shown (* $p \leq 0.05$; ** $p \leq 0.01$; *** $p \leq 0.001$; **** $p \leq 0.0001$). A two-way ANOVA model follow by Tukey post hoc analysis was used to determine the significance of time, genotype, or the interaction; n = 3–4 (A) and n = 3–6 (B); p-values are indicated. (C) Hematoxylin/Eosin staining of liver sections of mice maintained on the HFD. Two randomly selected microscopic fields are shown per genotype.

### The NCoRδ-/- mice did not display the dramatically enhanced resistance to diet-induced glucose intolerance seen in the NCoRω-/- mice

Diet-induced obesity in humans and in WT C57BL/6 mice leads to an impaired glucose tolerance that can progress to overt type 2 diabetes [47]. This was not observed in our NCoRω-/-

mice, which paradoxically display a strongly improved glucose tolerance compared to WT on either the LFD or HFD despite their enhanced adiposity and liver steatosis on the latter (Fig 4A, 4C and 4E) [25]. Notably this greatly enhanced resistance to diet-induced glucose intolerance in the NCoRω-/- mice was not observed in the NCoRδ-/- mice on either diet (Fig 4A, 4C and 4E). The NCoRδ-/- mice did however display an intermediate glucose tolerance phenotype on the HFD with a trend (p < = 0.11) toward slight improvement compared to WT (Fig 4C), as well as a tendency (p < = 0.37) toward a slightly enhanced insulin sensitivity (Fig 4B, as area under the curve in Fig 4D and 4F). Given insulin sensitivity for the NCoRω-/- animals was indistinguishable from WT despite their enhanced resistance to diet-induced glucose insensitivity (Fig 4B, 4D and 4F), it is possible that the NCoRδ knockout affects glucose utilization, but more modestly and through distinct mechanisms from those of the NCoRω-/- mutation (please see the Discussion).

## Ablation of the NCoRδ yields a unique pattern of hepatic gene expression distinct from that in WT or in NCoRω-/- mice

Changes in hepatic gene expression in the NCoRδ-/- and the NCoRω-/- mice compared to WT included genes that were changed in expression in one mutant genotype but not the other, changed in expression in the same direction in the two genotypes, or changed in opposite directions in the two different genotypes (Fig 5 and, more comprehensively, S5 Fig in S1 File and S1 Table). These results indicate that these corepressor splice variants recognize both different and common transcriptional targets.

Notably the changes in expression of sets of genes associated with hepatic steatosis and cholesterol metabolism exhibited largely opposite profiles in the NCoRδ-/- versus the NCoRω-/- livers (S4A and S4B Fig in S1 File), consistent with the opposite steatosis phenotypes of these mutants. The NCoRδ-/- mice also diverge from the NCoRω-/- mice in the extent of their resistance to HFD-induced glucose intolerance and exhibit a distinct hepatic gene expression profile in this regard as well (e.g. S4C Fig in S1 File).

A panel of nuclear receptors are known to play key roles in control of hepatic differentiation, homeostasis and metabolism, including Estrogen-Related Receptor-α (ERRα), Farnesoid X Receptors (FXRs), Glucocorticoid Receptors (GRs), Hepatic Nuclear Factor 4-α (HNF4α), Liver X Receptors (LXRs), Thyroid Hormone Receptors (TRs), Peroxisome Proliferator Activated Receptors α and γ (PPARα and γ), and Reverse Erb A (Rev-erb) [22, 48–50]. NCoRδ and NCoRω differ in their affinities for many of these nuclear receptors in vitro, including PPARα, PPARγ, TRα, FXR, and LXRα (the relative binding to NCoRδ versus NCoRω of these nuclear receptors by GST pulldown analysis [34] was 0.79 (95% C.I. 0.56–1.20), 0.88 (95% C.I. 0.66–1.30), 0.42 (95% C.I. 0.25–0.60), 0.35 (95% C.I. 0.29–0.43), 0.51 (95% C.I. 0.46–0.56), respectively (S3 Table). To investigate if the distinct hepatic transcriptomes of the NCoRδ-/- and NCoRω-/- mice might reflect, in part, the actions of these nuclear receptors we examined the relative expression of representative target genes in the livers of the two genotypes (Table 1 and S1 Table). Notably the expression of least several target genes for each of the nuclear receptors listed were altered, either up or down, in NCoRδ-/- vs. NCoRω-/- animals, with alterations in expression of PPARα and TR targets the most prominent. It should be noted that these changes in expression may represent either primary or secondary effects and several of these target genes are regulated by multiple nuclear receptors and non-receptor transcription factors.

Serum Response Element Binding factor 1 (SREBF1) and Carbohydrate Response Element Binding factor 1 (ChREBF1) also play essential roles in control of liver metabolism [51]; therefore, although not known to physically partner with NCoR, representative target genes for

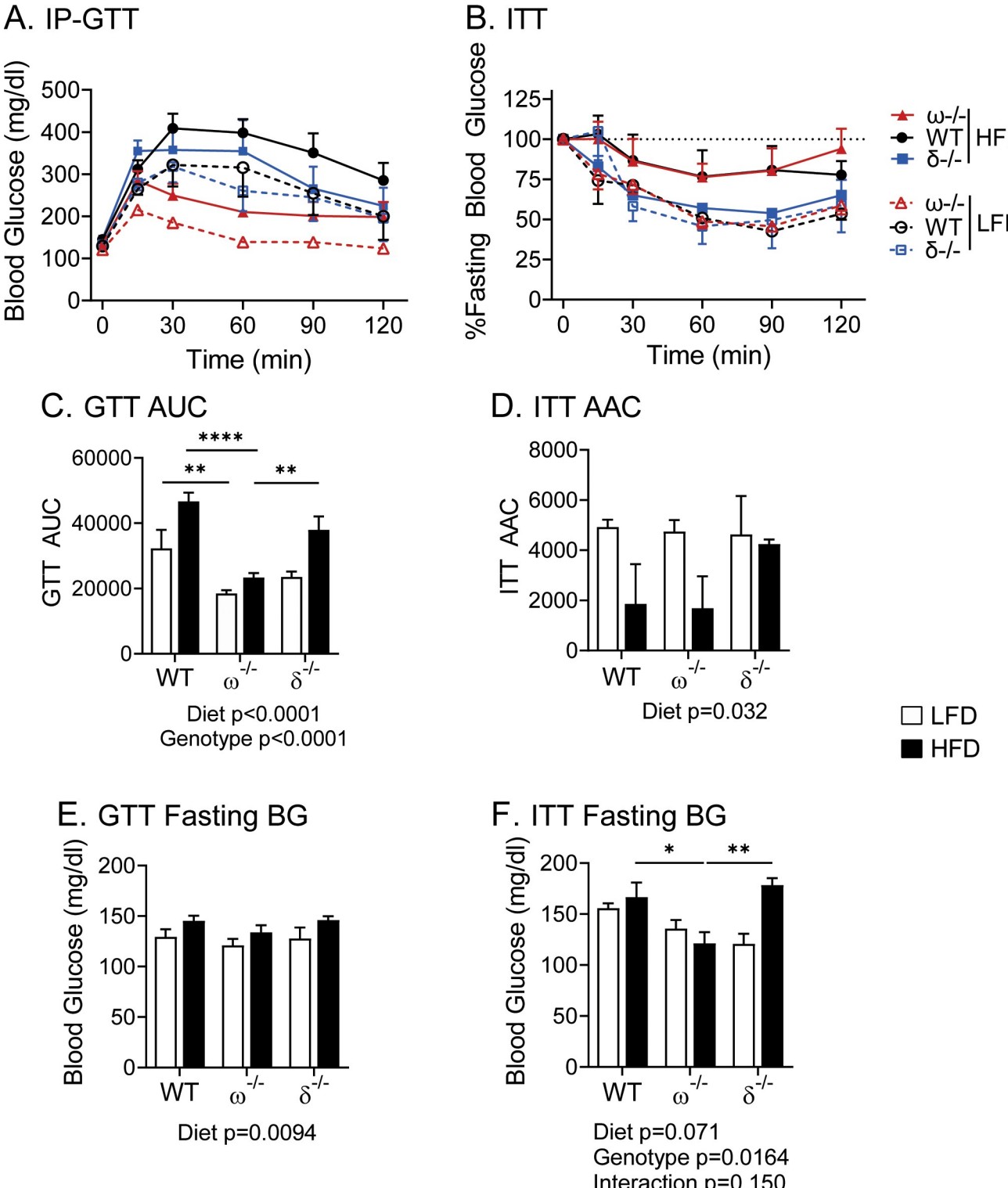

**Fig 4. Glucose and insulin tolerance in NCoRδ-/-, NCoRω-/-, and WT mice.** (A) Intraperitoneal-Glucose Tolerance Test (IP-GTT). (B) Insulin Tolerance Test (ITT). (C) Areas under the curve (AUC) for panel A. (D) Areas above the curve (AAC) for panel B. (E) Initial blood glucose levels for GTTs in fasted animals. (F) Initial blood glucose levels for ITTs fasted animals. Symbol and fill keys are to the right of the panels. Means and standard errors are indicated; p-values are indicated as in Fig 3. A two-way ANOVA model follow by Tukey post hoc analysis was used to determine the significance of time, genotype, or the interaction; n = 5–11 (A, C & E) and n = 3–7 (B, D & F); p-values are indicated.

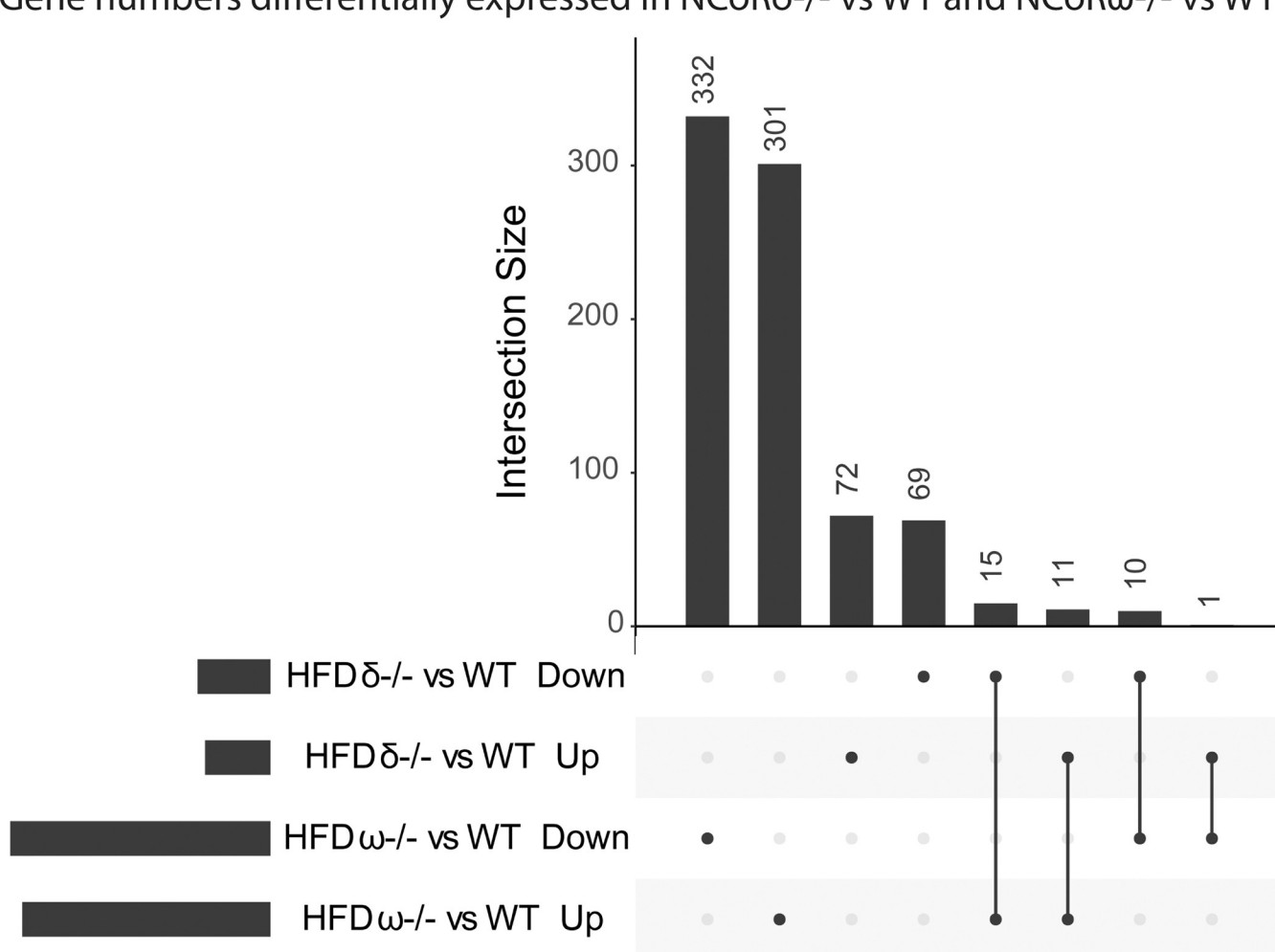

**Fig 5. Gene expression comparisons, UpSet plot.** Numbers of genes altered in hepatic expression in each mutant, up or down relative to WT, are depicted in the bar graph at the top of the panel. Dots below serve to link each bar to their label on the left. Vertical lines joining two dots indicate the bar above consists of genes sharing two labels, i.e. up in both NCoRδ-/- and NCoRω-/-.

these transcription factors are also included in Table 1. Notably expression of many of these SREBF1 and ChREBF1 target genes also differed in NCoRδ-/- vs. NCoRω-/- animals. Our results indicate that changes in NCoR alternative-splicing substantially and widely impact the receptor and non-receptor transcriptional regulatory networks known to control hepatic energy, glucose, and lipid metabolism.

## Gene Set Expression Analysis (GSEA) implicates specific biological pathways as mediators of the distinct NCoRδ-/- vs/ NCoRω-/- phenotypes

To more broadly elucidate the contributions of the observed transcriptional changes in the mutant mice to their phenotypes, and thus to better understand how shifts in the ratios of these splice variants alter hepatic gene expression in WT animals, we applied a GSEA KEGG pathway analysis. We focused first on identifying pathways that were altered differently in the

**Table 1. Relative expression of nuclear receptor/transcription factor target genes in NCoRδ-/- vs. NCoRω-/- livers.**

| | | | | | |
|---|---|---|---|---|---|
| **ERRα targets** | | | | | |
| ACO2 | MDH2 | OGDH | PFKP | G6PC | ACADM |
| COX5B | HK2 | GAPDH | FASN | PCK1 | PDK4 |
| ENO1 | PGK1 | SLC2A4 | IDH3A | CS | CYCS |
| FH1 | SDHA | SDHB | | | |
| **FXR targets** | | | | | |
| NROB2 | ABCB11 | ABCB11 | SLC51B | ABCC2 | ABCB4 |
| BAAT | SLC51A | CYP11A | APOC2 | LPL | PPARA |
| PCK1 | G6PC | SLC27A5 | | | |
| **GR targets** | | | | | |
| IRS1 | FKBP5 | PCK1 | ACER2 | GADD45B | BCL2L1 |
| CEBPD | SREBF1 | MCM2 | ADIPOR2 | AASS | FOXO1 |
| CDC40 | PPP1R3C | SLC7A2 | TAT | RS1 | CPT1A |
| LPIN2 | IDH1 | G6PC | IGFBP1 | CYP2E1 | |
| **HNF4α targets** | | | | | |
| CYP1A1 | APOA4 | DIO1 | OTC | TLR2 | BDH1 |
| SULT1B1 | EGR1 | CAR3 | CCND1 | MT2 | PDK4 |
| ALDOB | LY96 | CCR4 | GLYAT | | |
| **LXR targets** | | | | | |
| CYP7A1 | ABCA1 | AGXT | ABCD3 | THRSP | AGPAT2 |
| ABCG5 | LPL | EHHADH | FADS2 | ACACB | ABCA1 |
| NR0B2 | SREBF1 | FASN | SCD1 | | |
| **TR targets** | | | | | |
| ME1 | NCOA4 | FGF21 | CYP17A1 | PCK1 | FASN |
| DIO1 | GPD2 | NT5E | KCNK1 | BCL3 | PDK4 |
| THRSP | THRB | LDLR | SULT1A1 | | HR |
| **PPARα targets** | | | | | |
| FABP4 | ABCA1 | FADS2 | CYP7A1 | AQP3 | ACOT12 |
| ABAT | SCD1 | ACOT1 | ACOT4 | SLC25A10 | PLIN1 |
| ANGPTL4 | EHHADH | PCK1 | FABP2 | ADIPOR2 | CD36 |
| ACADM | DCIK1 | ACAA2 | SREBF1 | ACOT8 | LPIN2 |
| ABCD2 | PCK1 | CYP4F15 | ACACB | FADS2 | |
| **PPARγ targets** | | | | | |
| ACOX1 | PLI1 | FABP4 | CYP4A10 | ACACA | ACLY |
| CIDEC | SFN | FABP5 | SLC2a2 | FASN | PLIN2 |
| CEBPA | IGFBP1 | IL7 | CD36 | SCD1 | MOGAT1 |
| **REV-ERB targets** | | | | | |
| APOC3 | INSIG2 | G6PC | PCK1 | NFIL3 | CYP7A1 |
| PPARG | NR0B2 | NFIL3 | ARNTL | CD36 | LPL |
| NPAS2 | CPT1A | | | | |
| **SREBF1 targets** | | | | | |
| FASN | ACACA | SCD1 | ACSS2 | HMGCS1 | INSIG2 |
| HMGCR | PCSK9 | ME1 | MCC | ALDOC | CEBPD |
| SQLE | FABP4 | LPIN1 | GSK3A | LDLR | CDK5 |
| **ChREBP targets** | | | | | |
| SULT1E1 | RTP4 | SLC34A2 | LGALS3BP | EHHADH | CAV |
| CXCL1 | JUNB | HIF1A | ACACA | G6PC | FAS |

(*Continued*)

**Table 1.** (Continued)

| | | | | | |
|---|---|---|---|---|---|
| SCD1 | | | | | |

Representative target genes for each listed nuclear receptor/transcription factor are presented; most but not all have been demonstrated to be direct targets. A given target gene may be regulated by more than one nuclear receptor or transcription factor. Red = expression decreased in NCoRδ-/- vs. NCoRω-/-. Green = expression increased in NCoRδ-/- vs. NCoRω-/-. Black = expression statistically indistinguishable in the two genotypes. The corresponding quantitative data is provided in S1 Table and the expression of several of the key genes listed here was confirmed by qRT-PCR (S7 Fig in S1 File).

NCoRδ-/- mice versus the NCoRω-/- mice under the HFD (Fig 6). Many of these differences implicated at the pathway level, particularly the increased conversion of cholesterol to primary bile acids, decreased sterol and unsaturated fatty acid synthesis, decreased PPAR signaling (also seen in analysis of individual PPARα and γ targets in Table 1), and decreased inflammation in the NCoRδ-/- versus the NCoRω-/- mice, are highly consistent with, and likely contributory to, their distinct hepatic steatosis phenotypes.

Yet additional KEGG pathways were called out by the GSEA analysis in comparisons of NCoRδ-/- to WT or of NCoRω-/- to WT that were not called out in the comparison of mutant to mutant (S6 Fig in S1 File). Some of this lack of complete congruence among these comparisons likely reflects limits to the power of our analyses. However at least several of these pathways were altered from WT by both mutants in a statistically significant manner (e.g. S6 Fig in S1 File, color-coded green for shared upregulated and orange for shared downregulated pathways). These therefore likely represent pathways regulated in common by both the NCoRδ and NCoRω splice variants.

## Discussion

### Loss of NCoRδ resulted in reduced hepatic steatosis and reduced weight gain phenotypes compared to that seen in WT and/or on ablation of NCoRω

The alternative NCoR splicing investigated here either incorporates exon 37b+ to generate the NCoRω variant, or incorporates exon 37b- to generate the NCoRδ variant (Fig 1B; [25]). Differentiation of pre-adipocytes in culture is associated with a switch from production of predominantly NCoRω, which blocks adipogenesis when overexpressed, to predominantly NCoRδ, which promotes it. Consistent with these *ex vivo* results, genetic ablation of NCoRω while retaining NCoRδ expression in intact mice produced a greatly enhanced weight gain and adiposity on a HFD *in vivo* [35]. In the current manuscript we demonstrate that the NCoRδ-/- mice manifest the opposite phenotype, a suppressed overall weight gain compared to WT and the NCoRω-/- mice.

More relevant to the current manuscript, the NCoRδ-/- mice also exhibited an additional, strikingly divergent phenotype versus the NCoRω-/- animals: a greatly reduced hepatic steatosis under a HFD regimen. In comparison the NCoRδ-/- livers display relatively modest differences versus the WT animals (also observed in the transcriptional analysis, below). These results indicate that despite both splice variants being abundantly expressed in normal liver, the NCoRω variant exerts the more prominent role in this context. Taken together with our prior observation that the ratio of NCoRδ versus NCoRω in hepatocyte-derived cells in culture is itself responsive to nutritional inputs [52], we propose that alternative splicing of NCoR at exon 37 plays a key function in defining hepatic lipid accumulation and energy homeostasis, as well as serving to regulate other aspects of overall body weight and adiposity.

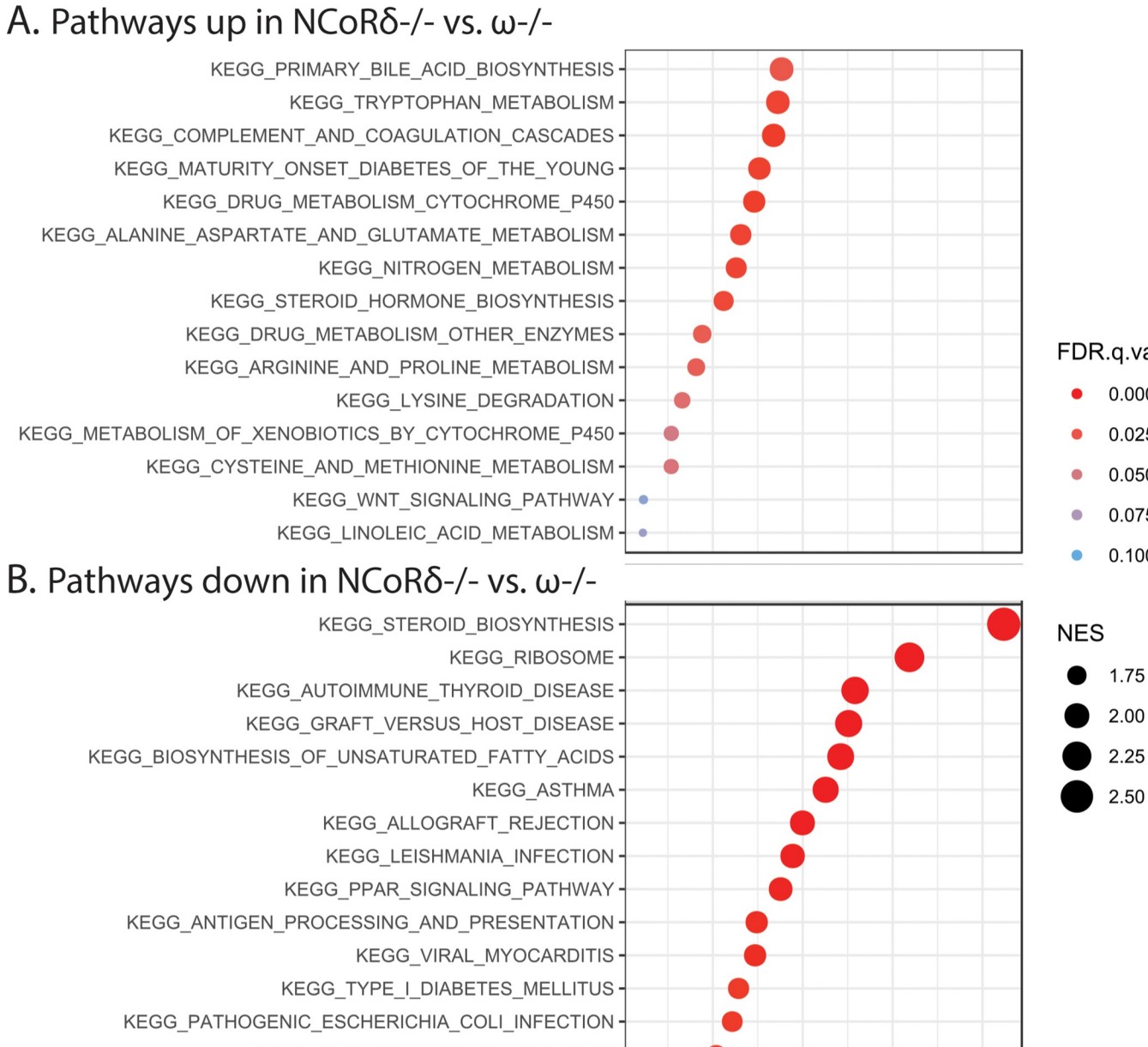

**Fig 6. GSEA analysis of KEGG pathways differentially regulated in NCoRδ-/- versus NCoRω-/- livers.** (A) Upregulated in NCoRδ-/- versus NCoRω-/-. (B) Downregulated in NCoRδ-/- versus NCoRω-/-. False Discovery Rate (FDR) and Normalized Enrichment Scores (NES) are indicated.

It is likely that some or many of the phenotypic effects we report here (or the transcriptional effects we discuss below) are not cell autonomous; some may be mediated instead by changes in hormonal or other forms of cell to cell or tissue to tissue signaling. Our experiments therefore serve to identify the overall actions of NCoRδ and NCoRω in the organism but do not presume to establish the tissue-specific origins of these actions. Similarly, certain effects of the

NCoR splicing mutants in the adult mouse may reflect the outcomes of events first established during earlier development.

## Transcriptome analysis identified genes that are regulated divergently, regulated by one but not by the other, regulated to different extents, or regulated indistinguishably by the two splice variants

RNA sequence analysis revealed different panels of hepatic genes in the NCoRδ-/- mice that relative to WT displayed changes in expression opposite to that of those in the NCoRω-/- mice, displayed changes in expression that were detected in one mutant genotype but not the other, or exhibited alterations in expression in the same direction in the two genotypes. The latter presumably include genes that are common targets of the two splice variants and thus contribute to their known overlapping biological functions. It is likely that the distinct transcriptomes mediated by the NCoRω and NCoRδ splice variants largely reflect their different affinities for different transcription factor partners (plus the secondary transcriptional changes mediated by these primary targets). Nonetheless it is possible that NCoRω and NCoRδ may also produce distinct transcriptional outputs by assembling distinct effector protein complexes once bound to the same transcription factor. For example, histone deacetylase 3 (HDAC3) is an important NCoR effector protein and liver-specific ablation of HDAC3 results in a hepatosteatotic/glucose tolerant phenotype resembling in many ways that of our NCoRω-/- mice [53]. Although both NCoRω and NCoRδ recruit HDAC3 *in vitro* we cannot rule out that differential recruitment of HDAC3 *in vivo* to one or more target genes also plays a role in one or more of the different actions of these splice variants [35].

Many of these changes in gene expression associated with the NCoRδ-/- versus the NCoRω-/- mice likely contribute to their divergent hepatic steatosis phenotypes; these include transcriptional changes indicative of increased conversion of cholesterol to primary bile acids, decreased steroid synthesis, decreased unsaturated fatty acid synthesis, and decreased inflammation. Yet other transcriptional alterations in a variety of additional energy, lipid, and carbohydrate pathways likely further contribute to the distinct steatosis phenotypes, including those involved in the metabolism of a variety of small metabolic intermediates and in amino acid utilization and degradation.

The changes to the hepatic transcriptomes of the NCoRδ-/- and NCoRω-/- mice included alterations in expression of genes that are known targets of the nuclear receptors that regulate important aspects of liver energy, carbohydrate, and lipid metabolism. Particularly prominent among the panel examined in this study were changes in the expression of target genes for PPARα and TR, as well as targets for the non-receptor transcription factor SREBPF1. PPARα is a key regulator of fatty acid β-oxidation and energy homeostasis [22, 48–50]. PPARα is a key regulator of fatty acid β-oxidation and energy homeostasis [22, 48]. TR regulates a broad series of biological processes throughout the organism but plays particularly important roles in liver in regulation of basal metabolic rate and lipid metabolism [22, 48]. SREBPF1 plays a central role in control of cholesterol and fatty acid lipogenesis [51].

The changes we observe in our panel of target genes are therefore generally consistent with prior observations and with the differences we observe in hepatic steatosis between the NCoRδ and NCoRω animals. Changes in TR and PPARα function (and that of LXRα and ERRα) have been implicated in the pro-steatotic hepatic effects of several prior pan-splice NCoR mutations (reviewed in [22]). Interestingly knockouts or dominant-negative mutations of TRα have been shown to decrease hepatic lipogenesis and triglyceride content and, in the former, to protect mice against high fat diet induced insulin resistance, whereas dominant negative mutations of TRβ are associated with increased hepatic lipogenesis, decreased hepatic β-oxidation, and

increased hepatic steatosis [54, 55]. Ablation of PPARα has been shown to be associated with elevated hepatic triglycerides, cholesterol esters, and steatosis [56]. ERRα knockout mice are protected against high fat diet-induced hepatic steatosis [57]. LXRα knockout mice develop an abundant cholesterol-laden liver steatosis on a high cholesterol diet yet display resistance to ob/ob mediated steatosis [58, 59]. It should be noted that nuclear receptors can mediate both repression and activation in a target gene-specific manner, making these comparisons of nuclear receptor versus corepressor knockouts complex to interpret.

Many genes involved in the regulation of hepatic metabolism are expressed cyclically under the control of the cellular clock [60, 61]. Important aspects of this circadian control are mediated by the recruitment of the NCoR/HDAC3 complex to these target genes through the cyclic binding of the Rev-erb nuclear receptor (either directly or through a tethered interaction with other DNA binding factors) [60, 61]. Disruption of this process leads to disruption of the circadian rhythm, increased lipogenesis and cholesterol synthesis, decreased fatty acid oxidation, and increased hepatic steatosis [22, 61]. Given the expression of at least several Rev-erb target genes are altered between the NCoRδ-/- and NCoRω-/- mice it will be interesting to examine if our splice-specific NCoR ablations influence this circadian rhythm/Rev-erb/gene expression phenomenon.

The question of whether the changes in liver steatosis in response to HFD-feeding are due to differences in the affinity of NCoRω and NCoRδ for specific nuclear receptors is difficult to answer conclusively. The NCoRω-/- and NCoRδ-/- mice ablate the expression of their respective isoform since the beginning of development in all tissues. As such, the phenotypic effects seen in any tissue are the integration of signals from multiple tissues resulting from the effects of direct targets of NCoR-nuclear receptor complexes and the regulatory cascades created by those complexes. Acknowledging these caveats, one plausibly direct mechanism for the increased steatosis observed in the livers of HFD-fed NCoRω-/- mice involves increased activity of LXRα resulting in increased expression of Srebf1. There is a growing appreciation that the magnitude of transcriptional response of a nuclear receptor, such as LXRα, is a function not only of the concentration of its ligand, but also the ratio of the coactivators and corepressors expressed within a cell [62, 63]. Thus, diverting expression of NCoR from an isoform with higher affinity for LXRα (NCoRω) to a lower affinity one (NCoRδ) should result in a less repressive/more activating transcriptional state for LXRα. Consistent with this, we observe a two-fold increase in expression of Srebf1 in the livers of NCoRω-/- mice compared to WT mice (S1 Table. p.adj = 0.01; 1.56-fold increase versus NCoRδ-/- mice, p.adj = 0.108). Increased activation of LXRα and expression of Srebf1 result in increased lipid synthesis and triglyceride metabolism and in hepatic steatosis [64, 65]. While this LXRα-mediated regulation of Srebf1 likely contributes to the increased fat deposition that we observe in the liver, the relative contribution of other nuclear receptors and other transcription factors, both within the liver and in other tissues, remains to be elucidated.

## The NCoRδ mice did not display the dramatically enhanced glucose tolerance exhibited by the NCoRω-/- mice

We also report here that the NCoRδ-/- mice fail to display the strongly improved glucose tolerance observed in the NCoRω-/- mice compared to WT. We previously proposed that the paradoxically enhanced glucose tolerance of the NCoRω-/- mice was due in part to a shift in hepatic gene expression away from gluconeogenesis to lipogenesis in the absence of altered insulin signaling [35]. Intriguingly the NCoRδ-/- mice exhibited a more generally pro-gluconeogenic hepatic gene expression signature, a phenomenon that may contribute to this difference. The HFD-fed NCoRδ knockout mice did nonetheless exhibit a "tendency" (p < = 0.11) toward a minimally improved glucose tolerance versus WT that, if confirmed, might be

explained by their corresponding tendency toward an enhanced insulin tolerance versus WT. Given that the NCoRω-/- animals did not display this alteration in insulin sensitivity, these results suggest that both NCoR splice variants may enhance glucose tolerance but to different extents and by different mechanisms, including potential contributions from non-hepatic tissues, such as pancreas or muscle.

### The effects of ablation of specific NCoR splice variants are distinguishable from the effects of ablation of all NCoR expression and of SMRT genetic manipulations

A *pan*-tissue knockout of all NCoR splice variants results in embryonic lethality [1]. The unimpaired viability of our NCoRδ-/- and NCoRω-/- animals therefore demonstrates that neither of these NCoR splice variants are individually essential in development or in the adult. Similarly certain tissue-specific *pan*-splice NCoR knockouts produce phenotypes, such as a strongly pro-oxidative signature in muscles [19, 35, 66], that were not fully recapitulated in either our NCoRδ-/- or our NCoRω-/- mice. This suggests that both NCoR variants must be ablated in muscle to produce the oxidative phenotype. In contrast the enhanced visceral adiposity associated with our NCoRω-/- mice resembles in certain aspects the phenotype reported for an adipocyte-specific *pan*-splice NCoR knockout [17], indicating that the loss of NCoRω variant alone is sufficient to produce the enhanced lipid accumulation observed by *pan*-NCoR ablation. Notably hepatocyte-specific, *pan*-splice NCoR knockouts produce a liver steatosis resembling that observed in our NCoRω-/- mice [67]; given this is not observed in our NCoRδ-/- animals it is likely that it is the specific loss of the NCoRω variant that also is responsible for this hepatic phenotype in the pan-splice knockout. We conclude that a given phenotype in a particular tissue may be mediated by NCoRδ, by NCoRω, or by both in a redundant fashion. We further suggest that a full understanding of the effects of experimental gene knockouts requires knowledge of the potentially divergent actions of the various splice variants in the wild-type animals.

The NCoR paralog SMRT is also expressed in liver as a series of alternatively-spliced variants that include or lack the analogous SMRT RID3 domain [68]. Although splice-specific knockouts of one or the other SMRT variant have not been reported, pan-splice ablation of SMRT in liver produces little or no detectable metabolic phenotype in mice [67]. These results taken together with ours suggest that NCoR, and its ω splice variant in particular, play the prevalent role in WT animals in this regard.

## Conclusions

Our results indicate that alternative splicing of NCoR at exon 37 serves a sensor and as a critical regulator of hepatic energy metabolism in mice and can exert diametrically opposite effects on specific target genes and on specific biological pathways. Yet other actions of the two NCoR splice variants do overlap indicating NCoRδ and NCoRω mediate redundant as well as splice-specific actions. We suggest that alternative mRNA splicing diversifies the properties of the resulting proteins to adapt NCoR function to the specific biological needs demanded in different tissues, at different stages of development, and by different metabolic and physiological states. Of note: the work reported here was performed in males and it will be valuable in the future to determine the parallels and differences exerted by these mutations in a female mouse background.

## Supporting information

**S1 File.**
(PDF)

**S1 Table. Relative gene expression in NCoRδ-/-. NCoRω-/-. and WT levels by RNAseq analysis.** An Excel workbook containing individual worksheets with normalized count data (FPKM) for each mouse sample and worksheets for each pairwise differential gene expression (DGE) top table. Indicated are the Mouse Genome Informatics (MGI) gene symbols, expression level base means, fold change (log base 2), log-fold-change standard error (LFC SE), Wald statistics, (stats), raw p values, and adjusted p values.
(XLSX)

**S2 Table. Oligonucleotide sequences used for genotyping and RT-PCR.** An Excel worksheet containing all of the oligonucleotide sequences used throughout this manuscript.
(XLSX)

**S3 Table. Quantification data of GST pulldown assays.** An Excel worksheet containing the quantification of GST pulldown assays between select nuclear receptors and NCoRω and NCoRδ RIDS [34]. Mean for NCoRω and NCoRδ refers to the mean percentage of the *in vitro* translated nuclear receptor copurified with GST-NCoRω or -NCoRδ RID fusion.
(XLSX)

**S1 Raw image.**
(TIF)

## Acknowledgments

We are deeply indebted to Liming Liu for superb technical help, and to the Mouse Biology Program and the Mouse Metabolic Phenotyping Center at the University of California at Davis for their analysis and excellent assistance. M.L.P. is also deeply grateful to his spouse, Susan R. Vigano, whose constant support contributed enormously to the completion of this work at a difficult time.

## Author Contributions

**Conceptualization:** Michael L. Goodson, Elsie L. Campbell, Chelsea A. Snyder, Briana M. Young, Martin L. Privalsky.

**Formal analysis:** Michael L. Goodson, Trina A. Knotts, Elsie L. Campbell, Chelsea A. Snyder, Briana M. Young, Martin L. Privalsky.

**Funding acquisition:** Martin L. Privalsky.

**Investigation:** Michael L. Goodson, Elsie L. Campbell, Chelsea A. Snyder, Briana M. Young, Martin L. Privalsky.

**Methodology:** Michael L. Goodson, Elsie L. Campbell, Chelsea A. Snyder, Briana M. Young, Martin L. Privalsky.

**Project administration:** Martin L. Privalsky.

**Resources:** Martin L. Privalsky.

**Software:** Trina A. Knotts.

**Supervision:** Michael L. Goodson, Martin L. Privalsky.

**Visualization:** Trina A. Knotts.

**Writing – original draft:** Michael L. Goodson, Martin L. Privalsky.

**Writing – review & editing:** Michael L. Goodson, Trina A. Knotts, Elsie L. Campbell, Chelsea A. Snyder, Briana M. Young.

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
