## [Decision Letter · Decision Letter 0]

8 Jul 2020

PONE-D-20-17204

Specific ablation of the NCoR corepressor delta splice-variant reveals alternative RNA splicing as a key regulator of hepatic metabolism.

PLOS ONE

Dear Dr. Goodson,

Thank you for submitting your manuscript to PLOS ONE. After careful consideration, we feel that it has merit but does not fully meet PLOS ONE’s publication criteria as it currently stands. Therefore, we invite you to submit a revised version of the manuscript that addresses the points raised during the review process.

The editors considered that your current manuscript would not achieve sufficient priority to merit acceptance. A number of considerations, in addition to reviewer comments, were weighed by the editors in reaching this decision. 

Please carefully address all of the reviewers' concerns and submit your revised manuscript by October 6th. If you will need more time than this to complete your revisions, please reply to this message or contact the journal office at plosone@plos.org. Please include the following items when submitting your revised manuscript:

We look forward to receiving your revised manuscript.

Kind regards,

Aijun Qiao, Ph.D.

Academic Editor

PLOS ONE

Journal Requirements:

Reviewers' comments:

Reviewer's Responses to Questions

**Comments to the Author**

1. Is the manuscript technically sound, and do the data support the conclusions?

Reviewer #1: Yes

Reviewer #2: No

2. Has the statistical analysis been performed appropriately and rigorously? 

Reviewer #1: Yes

Reviewer #2: No

3. Have the authors made all data underlying the findings in their manuscript fully available?

Reviewer #1: Yes

Reviewer #2: Yes

4. Is the manuscript presented in an intelligible fashion and written in standard English?

Reviewer #1: Yes

Reviewer #2: Yes

5. Review Comments to the Author

Reviewer #1: In this study, the authors investigated the effects on glucose and lipid metabolism of specific NCoR deficiency, not NCoR. Compared with the NCoR-/- mice, the NCoR-/- mice displayed the distinct phenotypes of reduced hepatic steatosis and reduced weight gain, but unable to resist to diet-induced glucose intolerance, different fromthe NCoR-/- animals. Furthermore, the authors compared the differential gene expression profiling between NCoR-/- and NCoR-/- mice by RNAseq and identified a series of hepatic changes in lipid, carbohydrate, and amino acid metabolic pathways that are likely to contribute to their distinct steatosis and glucose tolerance phenotypes. The results indicated that NCoR and NCoR variants exert both opposing and shared functions in many aspects of this phenomenon and in the organism as a whole. In general, the manuscript is an interesting article. However, there are some major concerns.

1. NCoR contains three C-terminal RIDs, NCoR only two, and here, the differential gene expression profiling between NCoR-/- and NCoR-/- mice had been identified by RNAseq, so, NCoR, as a nuclear Co-Repressor, could interact with different nuclear receptors and non-receptor transcriptional factors to repress gene expression. Please test the differential partner factors profiling between NCoR and NCoR by IP associated with MS.

2. Please confirm the results of RNAseq by qPCR, such as FASN, PDK4, FABP4, PCK1, etc.

Reviewer #2: Aim of the study was to characterize the functions of two alternative splice variants of the NCoR corepressor in the liver. The liver phenotypes of knockout mice for NCoR delta and NCoR omega variants (generated by the authors and published in JBC2011 with focus on adipose tissue) suggest that these isoforms have shared, distinct, and even opposing roles in regulating liver gene expression and pathways linked to metabolic liver phenotypes, including those induced by high-fat diet (steatosis, glucose intolerance). These complicated and difficult to dissect phenotypes can be understood from the distinct transcription factor (including nuclear receptor) - binding preferences of the two splice variants, from the different roles of the isoforms in affecting the entire corepressor complex (with NCoR being the major core subunit to assemble the liver complex), but also from the respective ratio of wild-type NCOR versus splice variants in the liver and in other tissues, including adipose tissue, that potentially trigger the liver alterations via tissue cross-talk.

The study addresses an important topic relevant for the better understanding of transcriptional networks governing liver and whole-organism physiology. Further, the study is both conceptually and experimentally challenging, as the function of the corepressor NCoR (gene names mouse Ncor1, human NCOR1) and its splicing variants is very complex and has both direct and indirect effects in multiple tissues and in gene expression pathways coordinated by a variety of transcription factor targets.

While it is a merit of the authors to take on this challenging task, the study in its current form is too preliminary, in part to poorly controlled, and mechanistically weak to justify publication.

Major concerns:

(1) The knockout mice are insufficiently characterised. The relative expression levels (mRNA, protein) of the NCoR WT, delta versus omega splice variants in WT and KO livers, and other tissues including adipose tissue, are not documented. The reader does not know which tissues are affected and what is the ratio between WT and splice variant. It is insufficient to refer to previous papers, the expression in the mice used for the current studies must be reported.

(2) The mouse experiments are statistically not sound, since the numbers of mice for each experiment are not indicated. This applies for RNA-seq as well, how many biological replicates were sequenced? It is even not indicated whether male or female mice were used.

(3) The interpretations of the diverse phenotypes are weak, e.g. direct effects of NCoR function in the liver (hepatocytes) are not distinguished from tissue crosstalk effects, for example adipose tissue (JBC2011) to the liver. Experiments using isolated hepatocytes from WT versus KO mice would help to distinguish these alternatives.

(4) Major conclusions must be derived from comparisons of WT vs NCoR delta KO and WT versus NCoR omega KO, not from comparing delta versus omega. Specifically, there seems no statistical difference between WT and NCoR delta KO mice on TG and liver weight increase upon HDF. As an example, the authors find ‘that NCoR delta -/- mice exhibited greatlyreduced hepatic steatosis as well as reduced overall weight gain on an HFD compared to the NCoR omega -/- mice. The NCoR delta -/- mice also fail to demonstrate the strong resistance to diet-induced glucose intolerance exhibited by the NCoR omega-/- animals.’ If they would have compared NCoR delta -/- mice to WT mice, the alternative interpretation would have been that the delta splicing variant has only moderate or no major functions in the liver.

(5) Fig. 3 A, B: Why does HFD feeding not effect liver weight and liver triglycerides (TG) in WT mice? This questions whether the mice respond appropriately. Again, there is no significant difference between WT and delta KO micde in weight and TG, so no liver phenotype of this splice variant.

(6) Fig. 4A, C, the omega -/-mice show clearly better glucose clearance than WT mice under both LFD and HFD, while there is no difference between omega -/-and WT mice in ITT test. How could omega -/- mice have no distinguishable insulin sensitivity from WT (line 232-233), but still can greatly enhance resistance to glucose intolerance (line 227-229)? In addition, how many mice were used in each group since some SD/SE are missing. Is there data significance at any timepoints between WT and omega -/-or delta -/-?

(7) Suggested is further to discuss better what we already know about the function of NCoR in the liver, without and within the entire corepressor complex (e.g. as cofactor for HDAC3, driving potentially many phenotypes), and which are its most likely nuclear receptor/transcription factor targets in the liver linked to the phenotypes (and target genes) of the splice variant knockouts. For example, how much do NCoR WT and splice variant KO overlap/oppose the TR beta and PPAR alpha KO changes in the liver?

(8) The writing accuracy has to be improved, there are errors in spelling and nomenclature. For example, in the Introduction the full names should be correctly for NCoR (nuclear receptor corepressor) and SMRT (silencing mediator of retinoic acid and thyroid hormone receptor). Suggested is also to include the official gene names mouse/human (Ncor1, NCOR1 and Ncor2, NCOR2). Line 53: bind to their nuclear receptor partners THROUGH (not thorough).

6. PLOS authors have the option to publish the peer review history of their article (what does this mean?). If published, this will include your full peer review and any attached files.

Reviewer #1: No

Reviewer #2: No

---

## [Author Response · Author response to Decision Letter 0]

17 Sep 2020

Editorial advice: 

This manuscript has been revised to adhere to PLOS ONE's style requirements.

2. PLOS ONE does not permit use of "data not shown." 

We have now either provided the data or, if the original observation was non-essential, we eliminated it from the revised manuscript.

In addition, we have deposited the results of the RNAseq analysis from this paper in the NCBI’s Gene Expression Omnibus and included the GEO Accession number (GSE157761; lines 168-169).

Reviewer #1 considered our manuscript to be an interesting article but expressed some notable concerns, specifically:

1. NCoR contains three C-terminal RIDs, NCoR only two, and here, the differential gene expression profiling between NCoRδ-/- and NCoRω-/- mice had been identified by RNAseq, so, NCoR, as a nuclear Co-Repressor, could interact with different nuclear receptors and non-receptor transcriptional factors to repress gene expression. Please test the differential partner factors profiling between NCoRδ and NCoRω by IP associated with MS.

We agree with the reviewer as to the value of defining the different partner interactions mediated by the two different corepressor splice variants. In response we have now provided the results of a series of GST-pulldown experiments. We employed the GST-pulldown approach in place of the IP/MS method for several reasons. Due to financial limitations we do not have any live colonies of our NCoR splice-specific mice. It would therefore require many months (and resources we do not have) to recover homozygotes from our frozen sperm samples as the first step to perform these experiments. More so, in our hands we have found the IP/LC/MS/MS approach itself requires substantial time and resources to perform (as well as representing a somewhat "fussy" method to quantitate reliably). We hope the reviewer will accept the GST-pulldowns, provided as data within the revised text (lines 295 to 299), to be an acceptable substitute. 

2. Please confirm the results of RNAseq by qPCR, such as FASN, PDK4, FABP4, PCK1, etc.

This information is now provided as the new Supplemental Fig. S7 (also please see line 375).

Reviewer #2 viewed the original as addressing an important topic relevant for the better understanding of transcriptional networks governing liver and whole-organism physiology. Although viewing it a merit to take on this "challenging task" reviewer #2 felt the study as originally submitted had flaws in its controls and mechanistic approaches. Specific concerns were:

(1) The knockout mice are insufficiently characterized. The relative expression levels (mRNA, protein) of the NCoR WT, delta versus omega splice variants in WT and KO livers, and other tissues including adipose tissue, are not documented. The reader does not know which tissues are affected and what is the ratio between WT and splice variant. It is insufficient to refer to previous papers, the expression in the mice used for the current studies must be reported.

We have now provided RT-PCR expression data for liver and a variety of additional tissues. In summary, in all tissues examined NCoRδ mRNA expression was undetectable in the NCoRδ-/- knockout and NCoRω mRNA expression was undetectable in the NCoRω-/- knockout (revised Supplemental Fig. S1B and line 198). We also confirmed the NCoR splice-variant mRNA levels from each of the liver samples used for the RNAseq analysis in Supplemental Fig. S8. 

We would have preferred to also provide protein expression levels. Unfortunately the NCoRδ versus NCoRω proteins can not be distinguished by electrophoresis/immunoblot. The problem arises from these splice variants migrating at essentially indistinguishable electrophoretic mobilities (due to their very similar large sizes, both over 240,000 kDa), and from the inability to generate NCoRδ-specific antibodies (NCoRδ differs from NCoRω due only to the loss of exon 37b+ sequencces; as a consequence there are no NCoRδ unique epitopes to which to raise antisera specific to this variant). Given the undetectable levels of NCoRδ and NCoRω mRNA in the respective knockouts, however, it is very unlikely that there is any significant corresponding protein expression in these animals. This is discussed in the revised text (lines 200-204). It should also be noted that these mice were generated to contain a germline (global) mutation that ablates the requried GT nucleotide sequence of the 5’ splice donor site for either the NCoRδ or the NCoRω splice-variant, rather than by using in vivo flox methodologies. Homozygosity each mutation was confirmed by genotyping in all mice. Conceptually NCoRδ-/- and NCoRω-/- mice should retain no residual WT NCoR loci in any tissue. 

(2) The mouse experiments are statistically not sound, since the numbers of mice for each experiment are not indicated. This applies for RNA-seq as well, how many biological replicates were sequenced? It is even not indicated whether male or female mice were used.

We now indicate that all the mice analyzed were males (lines 127 to 128) and provide "n" values for our experiments in the revised Figure Legends.

(3) The interpretations of the diverse phenotypes are weak, e.g. direct effects of NCoR function in the liver (hepatocytes) are not distinguished from tissue crosstalk effects, for example adipose tissue (JBC2011) to the liver. Experiments using isolated hepatocytes from WT versus KO mice would help to distinguish these alternatives.

Unfortunately we were forced to terminate our colonies of NCoR splice-specific knockout mice and it would take many months (and more resources than currently available to us) to recover homozygous animals from frozen sperm and to perform the hepatocyte experiment requested. We fully recognize that tissue autonomy is a valuable and interesting question but respectfully put forward that it was not the core question we were seeking to address in these, our first-step characterizations of the NCoRδ-/- animals. Our goal at this stage was to determine the overall NCoRδ-/- phenotype at the organismal level; we hope to have an opportunity to track down the specific tissue origins of each of the phenotypes in future follow-up experiments. Although a "Tu Quoque" defense, similar first reports of gene KOs have similarly often postponed analysis of autonomous versus non-autonomous tissue effects to later study. We fully appreciate the reviewer's point, however, and we now discuss this issue in the revised manuscript (lines 436 to 442). 

(4) Major conclusions must be derived from comparisons of WT vs NCoR delta KO and WT versus NCoR omega KO, not from comparing delta versus omega. Specifically, there seems no statistical difference between WT and NCoR delta KO mice on TG and liver weight increase upon HDF. As an example, the authors find ‘that NCoR delta -/- mice exhibited greatly reduced hepatic steatosis as well as reduced overall weight gain on an HFD compared to the NCoR omega -/- mice. The NCoR delta -/- mice also fail to demonstrate the strong resistance to diet-induced glucose intolerance exhibited by the NCoR omega-/- animals.’ If they would have compared NCoR delta -/- mice to WT mice, the alternative interpretation would have been that the delta splicing variant has only moderate or no major functions in the liver.

Our primary goal in this manuscript was to examine and contrast the splice-specific effects of NCoRδ and NCoRω; we therefore found it useful to present our narrative more in terms of the similarities and differences between the NCoRδ-/- and NCoRω-/- mice than in terms of comparisons of the mutants to WT. Nonetheless we do also present and discuss all of the corresponding mutant to WT comparisons for the benefit of readers seeking to examine this work from that perspective (e.g. lines 90-93, 206-208, 211-213, 229-234, 251-254, 271-275, 387-392, 403-407, 427 to 430).

We also note that although the effects of the NCoRδ-/- KO are generally modest compared to WT, the NCoRδ-/- KO nonetheless presents with a number of significant phenotypic differences relative to WT, including alterations in hematopoiesis, distinct body weight gains on low and high fat diets, and differences in liver gene expression profiles. Further, given NCoRδ and NCoRω are nearly equally expressed in WT liver and in many other tissues, determining the actions of both variants represents a key requirement for fully understanding the role of alternative NCoR splicing. The unexpectedly relatively modest hepatic effects of ablating NCoRδ, compared to the more dramatic effects of ablating NCoRω, was therefore important to establish. We have included and expanded our discussions of these issues in the revised manuscript (lines 206-208, 229-230, 251-258, 427-430).

(5) Fig. 3 A, B: Why does HFD feeding not effect liver weight and liver triglycerides (TG) in WT mice? This questions whether the mice respond appropriately. Again, there is no significant difference between WT and delta KO mice in weight and TG, so no liver phenotype of this splice variant.

We are unclear as to why we fail to see a more robust difference in liver weight and triglyceride levels between low fat and high fat diet-fed WT mice. There is a trend (p=0.079) toward increased liver weight with the HFD diet, but not a statistically robust one. This same phenomenon was also observed for WT animals in our previous NCoRω-/- study (representing a completely independent feeding trial) (35). We do note that there was a clear difference in hepatic steatosis between the low and high fat diets in the NCoRω-/- and NCoRδ-/- animals. We suspect that the extended time course (25 weeks) of our experiment and the formulation of the low fat control diet (Research Diets cat# D12450B; 35% kcal sucrose) available at the time of this study may be responsible for muting the HFD enhancement of steatosis anticipated for WT animals by increasing the steatosis in the LFD control mice. For example, it has been reported that chronic sucrose consumption in rats leads to increased hepatic steatosis (Souza Cruz EM et al., Long-term sucrose solution consumption causes metabolic alterations and affects hepatic oxidative stress in Wistar rats. Biol Open. 2020 Feb 28;9(3):bio047282. doi: 10.1242/bio.047282).

(6) Fig. 4A, C, the omega -/-mice show clearly better glucose clearance than WT mice under both LFD and HFD, while there is no difference between omega -/-and WT mice in ITT test. How could omega -/- mice have no distinguishable insulin sensitivity from WT (line 232-233), but still can greatly enhance resistance to glucose intolerance (line 227-229)? In addition, how many mice were used in each group since some SD/SE are missing. Is there data significance at any timepoints between WT and omega -/-or delta -/-?

We observed this same phenomenon previously and proposed that the unanticipated enhanced glucose tolerance of the NCoRω-/- mice (despite their elevated weights and adiposity) in the absence of altered insulin signaling was due in part to a shift in hepatic gene expression away from gluconeogenesis toward lipogenesis (35). Given the HFD fed NCoRδ knockout mice exhibited a “trend” toward a modestly improved ITT whereas the NCoRω-/- animals did not, these observations suggest that although both NCoR splice variants may enhance glucose tolerance they do so to different extents and likely by different mechanisms. We discuss these issues in the revised manuscript (lines 513-521).

We also now provide the "n" values requested in the figure legend. None of the individual time points in Figure 4B display statistic differences between the different genotypes; the integrated area above the curve (Figure 4D) proved the best means of visualizing and analyzing this data statistically. 

(7) Suggested is further to discuss better what we already know about the function of NCoR in the liver, without and within the entire corepressor complex (e.g. as cofactor for HDAC3, driving potentially many phenotypes), and which are its most likely nuclear receptor/transcription factor targets in the liver linked to the phenotypes (and target genes) of the splice variant knockouts. For example, how much do NCoR WT and splice variant KO overlap/oppose the TR beta and PPAR alpha KO changes in the liver?

We thank the review for this suggestion and provide an expanded discussion of these points in the revised manuscript (lines 482-496).

(8) The writing accuracy has to be improved, there are errors in spelling and nomenclature. For example, in the Introduction the full names should be correctly for NCoR (nuclear receptor corepressor) and SMRT (silencing mediator of retinoic acid and thyroid hormone receptor). Suggested is also to include the official gene names mouse/human (Ncor1, NCOR1 and Ncor2, NCOR2). Line 53: bind to their nuclear receptor partners THROUGH (not thorough).

Our apologies. We have corrected these errors and proofread the revised manuscript.

---

## [Decision Letter · Decision Letter 1]

25 Sep 2020

PONE-D-20-17204R1

Specific ablation of the NCoR corepressor δ splice-variant reveals alternative RNA splicing as a key regulator of hepatic metabolism.

PLOS ONE

Dear Dr. Goodson,

Thank you for submitting your manuscript to PLOS ONE. After careful consideration, we feel that it greatly improved but does not fully meet PLOS ONE’s publication criteria as it currently stands. Therefore, we invite you to submit a revised version of the manuscript that addresses those important points raised by the reviewer 1.

Please submit your revised manuscript within two months. If you will need more time than this to complete your revisions, please reply to this message or contact the journal office at plosone@plos.org. Please include the following items when submitting your revised manuscript:

We look forward to receiving your revised manuscript.

Kind regards,

Aijun Qiao, Ph.D.

Academic Editor

PLOS ONE

Reviewers' comments:

Reviewer's Responses to Questions

**Comments to the Author**

1. If the authors have adequately addressed your comments raised in a previous round of review and you feel that this manuscript is now acceptable for publication, you may indicate that here to bypass the “Comments to the Author” section, enter your conflict of interest statement in the “Confidential to Editor” section, and submit your "Accept" recommendation.

Reviewer #1: (No Response)

Reviewer #2: All comments have been addressed

2. Is the manuscript technically sound, and do the data support the conclusions?

Reviewer #1: (No Response)

Reviewer #2: Yes

3. Has the statistical analysis been performed appropriately and rigorously? 

Reviewer #1: (No Response)

Reviewer #2: Yes

4. Have the authors made all data underlying the findings in their manuscript fully available?

Reviewer #1: (No Response)

Reviewer #2: Yes

5. Is the manuscript presented in an intelligible fashion and written in standard English?

Reviewer #1: (No Response)

Reviewer #2: Yes

6. Review Comments to the Author

Reviewer #1: The authors have already stated the different partner interaction with the two different corepressor splice variants by GST-pull down experiments. Please discuss whether the changes in expression of sets of genes associated with hepatic steatosis and cholesterol metabolism exhibited largely opposite profiles in the NCoRδ-/- versus the NCoRω-/- livers were caused owing to different affinities of NCoRδ and NCoRω for many of nuclear receptors or other partners. Please provide the detailed method and results for GST-pull down in the manuscript.

Reviewer #2: (No Response)

7. PLOS authors have the option to publish the peer review history of their article (what does this mean?). If published, this will include your full peer review and any attached files.

Reviewer #1: No

Reviewer #2: No

---

## [Author Response · Author response to Decision Letter 1]

2 Oct 2020

Authors' response to reviewers' comments on PONE-D-20-17204-R1:

Again, we sincerely thank the reviewers and the editor for their efforts on our behalf. We believe that their suggestions have improved our manuscript. We hope that we have adequately answered their concern and that the revised manuscript is now acceptable for publication. We detail our response below:

Reviewer #1 requested that we discuss whether the largely opposing effects we observe with the ablation of NCoRδ and NCoRω on steatosis and cholesterol metabolism can be related to the differences in affinity for various nuclear receptors between NCoRδ and NCoRω that are observed in vitro. Reviewer 1 also requested that we provide more detailed data and methods for the GST pulldown analysis of corepressor isoform-nuclear receptor interactions. :

1. The authors have already stated the different partner interaction with the two different corepressor splice variants by GST-pull down experiments. Please discuss whether the changes in expression of sets of genes associated with hepatic steatosis and cholesterol metabolism exhibited largely opposite profiles in the NCoRδ-/- versus the NCoRω-/- livers were caused owing to different affinities of NCoRδ and NCoRω for many of nuclear receptors or other partners. Please provide the detailed method and results for GST-pull down in the manuscript.

We now discuss one possible mechanism by which the differences we observe in affinity between NCoRδ and NCoRω for LXRα could lead to changes in hepatic steatosis and cholesterol metabolism through changes in Srebf1 expression (lines 539-558). While this mechanism very likely contributes to the enhanced hepatic steatosis in the livers of the NCoRω-/- mice that is not observed in the livers of the NCoRδ-/- mice, it also seems very likely that other nuclear receptors and transcription factors, both within the liver and in other tissues, likely also contribute to the metabolic phenotypes we observe in the liver (lines 513-527).

Quantification data from the GST pulldown assays used to calculate the affinity ratios (lines 325-329) for NCoRδ:NCoRω in the text of manuscript are now provided in Table S3. A detailed method for the GST pulldown assays is also provided (lines 148-177).

---

## [Editor Report · Decision Letter 2]

12 Oct 2020

Specific ablation of the NCoR corepressor δ splice-variant reveals alternative RNA splicing as a key regulator of hepatic metabolism.

PONE-D-20-17204R2

Dear Dr. Goodson,

We’re pleased to inform you that your manuscript has been judged scientifically suitable for publication and will be formally accepted for publication once it meets all outstanding technical requirements.

Kind regards,

Aijun Qiao, Ph.D.

Academic Editor

PLOS ONE

---

## [Editor Report · Acceptance letter]

15 Oct 2020

PONE-D-20-17204R2 

Specific ablation of the NCoR corepressor δ splice variant reveals alternative RNA splicing as a key regulator of hepatic metabolism. 

Dear Dr. Goodson:

I'm pleased to inform you that your manuscript has been deemed suitable for publication in PLOS ONE. Congratulations! Your manuscript is now with our production department. 

Kind regards, 

on behalf of

Dr. Aijun Qiao 

Academic Editor

PLOS ONE